# Food-Waste Valorisation: Synergistic Effects of Enabling Technologies and Eutectic Solvents on the Recovery of Bioactives from Violet Potato Peels

**DOI:** 10.3390/foods12112214

**Published:** 2023-05-31

**Authors:** Giorgio Grillo, Silvia Tabasso, Giorgio Capaldi, Kristina Radosevic, Ivana Radojčić-Redovniković, Veronika Gunjević, Emanuela Calcio Gaudino, Giancarlo Cravotto

**Affiliations:** 1Department of Drug Science and Technology, University of Turin, Via P. Giuria 9, 10125 Turin, Italy; giorgio.grillo@unito.it (G.G.); silvia.tabasso@unito.it (S.T.); giorgio.capaldi@unito.it (G.C.); vgunjevic@agr.hr (V.G.); giancarlo.cravotto@unito.it (G.C.); 2Faculty of Food Technology and Biotechnology, University of Zagreb, Pierottijeva 6, 10 000 Zagreb, Croatia; kristina.radosevic@pbf.unizg.hr (K.R.); irredovnikovic@pbf.hr (I.R.-R.); 3Faculty of Agriculture, University of Zagreb, Svetošimunska Cesta 25, 10 000 Zagreb, Croatia

**Keywords:** *Solanum tuberosum*, potato peel valorisation, antioxidant activity, phenolic compounds, green extraction, ultrasound-assisted extraction, microwave-assisted extraction, shelf life

## Abstract

The recovery of valuable bioactive compounds from the main underutilised by-products of the food industry is one of the greatest challenges to be addressed in circular economy. Potato peels are the largest waste generated during potato processing. However, they could be a potential source of valuable bioactive compounds, such as polyphenols, that can be reused as natural antioxidants. Currently, environmentally benign enabling technologies and new types of non-toxic organic solvents for the extraction of bioactive compounds may dramatically improve the sustainability of these processes. This paper focuses on the potential inherent in the valorisation of violet potato peels (VPPs) by recovering antioxidants using natural deep eutectic solvents (NaDES) under ultrasound (US)- and microwave (MW)-assisted extraction. Both the enabling technologies provided performances that were superior to those of conventional extractions in terms of antioxidant activity determined by the DPPH· (2,2-diphenyl-1-picrylhydrazyl) assay. In particular, the most promising approach using NaDES is proven to be the acoustic cavitation with a Trolox eq. of 1874.0 mmol_TE_/g_Extr_ (40 °C, 500 W, 30 min), vs. the 510.1 mmol_TE_/g_Extr_ of hydroalcoholic extraction (80 °C, 4 h). The shelf-life of both hydroalcoholic and NaDES-VPPs extracts have been assessed over a period of 24 months, and found that NaDES granted a 5.6-fold shelf-life extension. Finally, the antiproliferative activity of both hydroalcoholic and NaDES-VPPs extracts was evaluated in vitro using the MTS assay on human tumour Caco-2 cells and normal human keratinocyte cells (HaCaT). In particular, NaDES-VPPs extracts exhibited a significantly more pronounced antiproliferative activity compared to the ethanolic extracts without a noteworthy difference between effects on the two cell lines.

## 1. Introduction

Potato (*Solanum tuberosum*) is one of the most widely grown vegetables in the world and the fourth largest crop after rice, wheat, and corn, according to the 2021 FAO estimates [1]. The potato processing industry generates large quantities of by-products, mainly potato peels (PPs), that pose a disposal problem for the potato industry as wet wastes are a source of plant spoilage and pathogenic infections [2,3,4].

However, in a circular approach to the green economy, these by-products have recently been exploited as livestock feed or fertiliser [5,6,7,8] and as a new resource to produce biofuels and biogas [9,10,11]. Moreover, in line with the concept of “One Health”, potato waste could represent an even more affordable and effective feedstock for the extraction of value-added compounds, such as dietary fibres, natural antioxidants, biopolymers, and natural food additives, in a circular perspective [12,13,14]. PPs contain a wide range of nutritionally interesting components, such as phenolic compounds (chlorogenic acids, flavonoids, etc.) [15], glycoalkaloids [16], cell wall polysaccharides, and dietary fibre [17], suggesting that this by-product could have a wide range of potential re-uses [18,19,20]. Moreover, some red- or purple-coloured genotypes, such as the Siècle or Vitelote varieties, have been found to contain high levels of anthocyanins [18,21,22]. Phenolic compounds are well known for their health-promoting activities, such as antioxidant and anti-inflammatory activity [20,23]. Conventional solid/liquid extractions (SLE) remain the most common method for extracting bioactive compounds and, in particular, polyphenols from PPs [18,19]. However, conventional extraction processes heavily rely on the use of organic solvents, long processing times, high temperatures, and high energy consumption, which can have a negative impact on both human health and the environment. To overcome these constraints, innovative processes for the recovery of polyphenols from agro-industrial wastes have been proposed and thoroughly investigated [24,25,26].

The most promising innovative extraction techniques documented in the literature include Microwave-Assisted Extraction (MAE) [27,28], Ultrasound-Assisted Extraction (UAE) [29,30], Supercritical Fluid Extraction (SFE) [31], Pulsed Electric Fields (PEF) [32,33], Pressurised-Liquid Extraction (PLE), and Subcritical Water Extraction (SWE) [34,35].

Among others, MAE is an eco-friendly technique capable of penetrating the vegetable matrix and interacting with polar components. MW allows for fast solvent heating and a high extraction efficiency in short times, as documented in the pioneering work of Singh et al. (2014) for the extraction of polyphenols from PPs [36]. Likewise, UAE is a versatile, flexible, and simple technique that requires relatively low capital investment and is scalable for commercial use [37]. It amplifies extraction by accelerating diffusion phenomena and improving solvent penetration and mass transfer. UAE has been demonstrated significant improvements in the recovery of polyphenol extracts from PPs, compared to conventional extraction methods, as reported by Kumari et al. (2017) [38]. 

Moreover, the use of green and sustainable solvents, in conjunction with the application of the abovementioned environmentally friendly technologies, represents a promising holistic approach for the development of “green” extraction processes within the United Nations development plan based on 17 sustainable goals (SDG) [39,40]. In this framework, the agri-food industries have strived to improve their process sustainability through strategies based on the implementation of emerging sustainable technologies with the use of green solvents to decrease the overall environmental impact and improve the process efficiency for transforming agri-food residues into high-value-added products [41,42].

In particular, replacing harmful solvents with more environmentally friendly alternatives is not trivial and, in some cases, new challenges and limitations may arise due to the different physicochemical properties of the new solvents under consideration [37]. Several alternatives have recently been introduced in the field of sustainable extraction, such as supercritical fluids, neoteric, bio-based, and supramolecular solvents [43].

Among the neoteric ones, deep eutectic solvents (DESs), including those of natural origin (NaDESs), composed of plant metabolites [44,45], have recently received the highest attention thanks to their low toxicity, biocompatibility, and recyclability [46,47]. Natural Deep Eutectic Solvents (NaDES) are eutectic mixtures usually produced by the complexation of a quaternary ammonium salt (a hydrogen bond acceptor (HBA) with a hydrogen bond donor (HBD). The most widely used HBA constituent is choline chloride (ChCl), an inexpensive salt, while the most commonly used HBDs are alcohols, carboxylic acids, sugars, amino acids, and urea [48]. A special advantage in using NaDES for the extraction of bioactive from residual biomass is their ability to permeate and modify biomass cell walls and tissues and facilitate the release of compounds [49]. NaDESs have shown great potential for emerging green extraction technologies, especially when coupled with dielectric heating in MAE, due to their capability to interact with the electromagnetic field of MW [50]. Moreover, NaDES are expected to be widely transferred to industry in the coming years [51].

The present study aimed to investigate the potential recovery of polyphenols from Violet PPs waste using MW and US as enabling extraction technologies in combination with sustainable solvents. The NaDES chosen for this work is the ChLA, an equimolar mixture of ChCl and lactic acid (LA). The antioxidant activity of the VPPs extracts was evaluated using the DPPH· scavenging activity (2,2-diphenyl-1-picrylhydrazyl assay). The results have been compared with those of conventional extractions, and the shelf-life of both hydroalcoholic and NaDES-PPs extracts has been evaluated over a period of 24 months. Finally, the biological activity of the NaDES-PPs extract was investigated in terms of its antiproliferative activity adopting the so-called MTS assay, both on human tumour Caco-2 cells and human skin HaCaT cells. 

## 2. Materials and Methods

### 2.1. Biomass Material and Chemicals

The *Solanum tuberosum* L. cv Vitelotte peels used in this work were bought at the city market (Turin, Italy) and come from a biological culture. Before use, violet potato peels (VPP) were freeze-dried and milled using a laboratory blender (HGBTWTS360, Waring Blender, Stamford, USA). Sieving was applied to select <1000 µm granulometry (Giuliani, Turin, Italy). 

All chemicals were purchased from Sigma-Aldrich and used without further purification. NaDES was obtained via heating: ChLA was prepared with equimolar ratios of choline chloride (ChCl) and lactic acid (LA) [52]. The two components were stirred and heated at 50 °C in a Xelsius reactor (LabTech, Bergamo, Italy) without adding water until a homogeneous liquid was formed. ChLA was finally collected for biomass extraction without further purification. 

### 2.2. Conventional Extraction (Hydroalcoholic Extraction)

For the sake of comparison, conventional reflux extraction was performed with an EtOH hydroalcoholic solution. The result of this test was used as a benchmark [53]. In a typical extraction, 10 g of dry VPP (previously milled) was mixed inside a round-bottom flask with the correct amount of EtOH or hydroalcoholic solution at the 1:20 S/L ratio. The mixture was continuously stirred while reflux conditions were reached by means of an oil bath. After extraction, the solutions were vacuum filtered (25 μm filter paper pore size), and the matrices were thoroughly washed with fresh extraction solvent. The alcoholic fraction was removed using a rotary evaporator, and the crude extracts were then freeze-dried (LyoQuest–85, Telstar, Barcelona, Spain), and the dry material was analysed for antioxidant activity by means of DPPH· assay. For sake of comparison, the same procedure was applied, and the hydroalcoholic solution was replaced with ChLA. All the other parameters were kept unchanged. ChLA solutions were analysed for antioxidant activity. Every test was performed in triplicate, and results are reported as average value ± the standard deviation (S.D.).

### 2.3. Microwave-Assisted Extraction (MAE)

MAE was performed in a SynthWAVE reactor (Milestone Srl, Bergamo, Italy), a pressurisable multimode MW system that can work under an inert atmosphere (N_2_). Tests were performed by mixing 1 g of dry VPP (previously milled) with the desired solvent at the 1:20 S/L ratio. The protocol was applied to different solvent systems, namely, hydroalcoholic solutions (70:30 EtOH/H_2_O) and ChLA NaDES. Before each run, the system was purged with nitrogen three times to reduce oxygen-derived degradations. The reactor was finally pressurised with 5 bars of N_2_ to avoid solvent evaporation at the working temperature. All tests were performed at a maximum power of 1500 W of irradiation with a heating ramp of 5 min. The extractions have been carried out at different temperatures (80, 100, and 120 °C) and times (60, 30, and 5 min) with 650 rpm of magnetic stirring. The latter allows to avoid the generation of eventual hotspots in the extraction media due to MW irradiation. After the extraction, the solutions were vacuum filtered (25 μm filter paper pore size), and the matrices were thoroughly washed with fresh extraction solvent. Where necessary, the alcoholic fraction was removed by a rotary evaporator, and the crude extracts were then freeze-dried (LyoQuest–85, Telstar, Barcelona, Spain), and the dry material was analysed for antioxidant activity by means of DPPH· assay. ChLA solutions were analysed for antioxidant activity by means of DPPH· assay as such. Every test was performed in triplicate, and results are reported as average value.

### 2.4. Ultrasound-Assisted Extraction (UAE)

UAE extractions were performed using an immersion sonotrode (HNG-20500-SP, Hainertec, Suzhou, China), working at a frequency of 21 kHz, with 100 W and 500 W exploited during the screening. Extractions were carried out by mixing 5 g of dry VPP (previously milled) with the desired solvent at the 1:20 S/L ratio. The mixture was placed in a Pyrex^®^ thimble and cooled by means of an ice bath. The temperature was measured throughout UAE and was maintained approx. at 40 °C to preserve cavitation efficiency [54]. The abovementioned protocol was applied to different solvent systems: hydroalcoholic solution (70:30 EtOH/H_2_O ratio) and ChLA NaDES. All the extraction performed with the US did not require stirring due to the peculiar mass-transfer enhancement of cavitation phenomena. After extraction, the solutions were vacuum filtered (25 μm filter paper pore size), and the matrices were thoroughly washed with fresh extraction solvent. Where necessary, the alcoholic fraction was removed using a rotary evaporator, the crude extracts were then freeze-dried (LyoQuest–85, Telstar, Barcelona, Spain) and the dry material was analysed for antioxidant activity by means of DPPH· assay. ChLA solutions were analysed for antioxidant activity by means of DPPH· assay. Every test was performed in triplicate, and results are reported as average value ± the S.D.

### 2.5. Antioxidant Activity—DPPH·Assay

The antioxidant activity of the extracts was evaluated following the method described by Brand-Williams et al. by using the stable free radical DPPH· (2,2-diphenyl-1-picrilidrazile) [55]. The DPPH· radical inhibition, caused by the VPP extracts and measured by the decolouration of the solution (from violet to colourless), was monitored and referred to a Trolox methanolic solution, considered an antioxidant standard. The EC50 (the extract concentration able to inhibit 50% of the DPPH· radical at equilibrium) was evaluated as the scavenging activity parameter. Different concentration solutions of the dry extracts were prepared by operating subsequent dilutions, and the absorbance was read at 515 nm (Cary 60 UV-vis spectrophotometer, Agilent Technologies, Santa Clara, CA, USA). Bobo Least Squares software (ver. 0.9.1.) was used to process the absorbance data obtained to define a proper Probit regression [56]. A blank containing only water and methanol was used to zero the instrument; a blank sample containing the dry extract, without the DPPH· radical, was used to evaluate the matrix effect; and a reference sample containing water and DPPH· radical was used to normalise the results and verify the reactive absorbance.

### 2.6. Cell Proliferation Assay

Cytotoxicity and antiproliferative activity were evaluated in vitro using the CellTiter 96^®^ AQ_ueous_ One Solution Cell Proliferation assay (briefly, MTS assay). Two human adherent cell lines were used for this test: cancer Caco-2 cells derived from the colorectal adenocarcinoma and normal human keratinocyte cells (HaCaT). The caco-2 cell line was cultivated in DMEM supplemented with 20% (*v*/*v*) FBS and 1% (*v*/*v*) antibiotic/antimitotic solution, and the HaCaT cell line was cultivated in DMEM supplemented with 5% (*v*/*v*) FBS and 1% (*v*/*v*) antibiotic/antimitotic solution. The cell lines were kept in BioLite Petri dishes (Thermo Fisher Scientific, Waltham, MA, USA) in an incubator with a humidified atmosphere and 5% *v*/*v* CO_2_ at 37 °C. Single tests on the antiproliferative activity and cytotoxicity of the extracts were performed in 96-well plates (Thermo Fisher Scientific, USA) that had been seeded with exponentially growing cells at an initial concentration of 3 × 10^4^ cells per well in 100 μL of culture media. After 24 h of incubation, under cell cultivation conditions, the cells were treated with the extracts. The raw VPP were all diluted in the culture medium, then applied to the cells, resulting in final volume ratios of 0.5%, 2%, and 5% (*v*/*v*). Treatment lasted for 72 h in the incubator, followed by the MTS assay. The assay was carried out according to the manufacturer’s instructions with a few modifications. A 10 μL volume of MTS reagent was added to each well, and the cells were incubated for 3 h; the absorbance was measured at 492 nm on the microplate reader (Tecan, Männedorf, Switzerland). Cell viability percentage was expressed as the ratio between the absorbances of the treated versus nontreated control cells. The tests were performed in triplicate with four parallels for each volume ratio.

### 2.7. DCF-DA Assay

Reactive oxygen species (ROS) formation was determined spectrofluorimetrically by DCF-DA assay. HaCaT and Caco-2 cells were seeded in 96-well black plates at an initial concentration of 1 × 10^5^ cells/mL and incubated for 24 h. The next day, cells were treated with VPP extracts (5%, *v*/*v*) for a further 20 h. Cellular oxidation was induced by adding 100 μM H_2_O_2_ and incubating for 4 h. Cells were washed with PBS, and 100 μL of 50 μM DCF-DA was added to each well. Plates were incubated for 30 min in the dark and subsequently read by spectrofluorometer (Carry Eclipse, Varian, Palo Alto, CA, USA) at λex = 485 ± 10 nm and λem = 530 ± 12 nm.

## 3. Results and Discussion

### 3.1. Hydroalcoholic Extraction

Hydroalcoholic solutions are usually considered as the elective solvent for polyphenols extraction due to their polarity and the chance to recover the ethanolic fraction by means of distillation.

#### 3.1.1. Conventional Protocol—Hydroalcoholic Solution

In this first section, we screened different water/ethanol ratios, aiming to determine which one ensures the best performance in terms of dry yields. The latter was defined by solid residue quantification after solvent removal. In particular, according to the following equation: [(Dry Extract Weight)/(Matrix Weight)] × 100.

The tests were conducted with a conventional protocol (see Section 2.2) to evaluate only the effect owed to solvent composition. Results defined the 70:30 ratio as the most promising one, with an overall yield of 15.67% versus 5.69% and 10.53% of 99.8% and 50:50, respectively. The best extraction media was then selected for further screenings. The dry yield determination was selected as a preliminary test with the intention of defining a valorisation protocol that focuses not only on the extract activity but also on the process productivity, paving the way to a suitable valorisation protocol of a largely produced food residue.

To better understand the nature of the recoverable extract and its robustness, the conventional extraction was exploited, extending to 4 h for the procedure, investigating the antioxidant activity at RT and 80 °C. The results, reported in Table 1 and achieved with the optimised hydroalcoholic solution, are compared to the 1 h extraction.

As partially expected, hot extraction with prolonged time affects the overall activity of the extract (161.1 vs. 339.3 µmol_TE_/g_Extr_, after 4 and 1 h, respectively), whilst low temperature could preserve this feature also after 4 h. This information confirms the thermolabile nature of the VPP bioactive fraction.

#### 3.1.2. Technology Screening—Hydroalcoholic Solution

Starting from the gathered information, two of the main enabling technologies, namely, MW and US, were tested by exploiting the optimised EtOH:H_2_O 70:30 solution. According to the nature of the cavitation phenomena, behaving well at temperatures far from the solvent boiling point [54], the US-assisted samples have been collected at a temperature of ca. 40 °C. Thus, a comparison between conventional and MW-assisted protocols can only be indirect. For the latter, we adopted the lower extraction temperature, ascribable to the conventional extraction (Table 2).

It is possible to hypothesise a different selectivity in the quality of the extract, where, apparently, the MW enhance the recovery of non-active metabolites, according to the DPPH· test. On the other hand, UAE appears to be the less effective method, with limited activity and yield. Hence, the conventional approach has been adopted as a reference benchmark for further tests since it leads to the best outcome.

With the aim to better exploit the key features of the enabling technologies addressed in this work, additional investigations with the hydroalcoholic system have been devoted to further characterise their behaviour.

#### 3.1.3. US-Assisted Extraction (UAE)—Hydroalcoholic Solvent

To enhance the poor outcome achieved by means of UAE, extraction time and cavitation power have been evaluated, extending the first from 30 to 60 min and increasing the latter from 100 W to 500 W (See Table 3).

The results confirm that VPP extraction with acoustic cavitation is affected by some drawbacks from both dry yield and antioxidant activity points of view. Likely, stronger or prolonged treatments result in the degradation of metabolites, leading to condensation products with poor solubility. This behaviour, as an example, is typical of anthocyanins compounds [57]. This phenomenon is boosted by the presence of atmospheric oxygen in the extraction open vessel, as well. For this reason, attention has been moved to the MW-assisted procedure.

#### 3.1.4. MW-Assisted Extraction (MAE)—Hydroalcoholic Solvent

Starting from the previous results (see Table 2), the VPP MAE screening with hydroalcoholic solvent was pursued by increasing the extraction time (60 min vs. 30 min, at 80 °C) and investigating a higher temperature for shorter extraction time (5 min, 120 °C), as well. An intermediate protocol was adopted as a control (30 min, 100 °C) (see Table 4).

According to the results reported in Table 4, it is possible to state that the milder treatment displays lower selectivity, with high extraction yield and low activity. On the other hand, the flash protocol at 120 °C results in the best outcome with the highest Trolox equivalents (337.1 µmol_TE_/g_Extr_), approximately matching the value of the conventional protocol, adopted as a benchmark (339.3 µmol_TE_/g_Extr_).

### 3.2. ChLA Extraction

The second section of this work is dedicated to exploring VPP extraction by means of the NaDES system, more precisely with ChLA. This choline chloride-based eutectic solvent has been selected due to previous experiences with polyphenols extraction, in addition to their stabilisation and biological activity [50]. For the sake of comparison, the screening followed the procedure adopted for the hydroalcoholic protocol. Hence, ChLA has been studied with conventional and unconventional techniques (UAE and MAE). It is worth noting that no dry yields are reported hereafter since the aim of this study is to explore the activity of formulates built by the combination of extract/NaDES as reported by several studies [58,59,60]. The isolation of metabolites recovered by means of NaDES usually requires the exploitation of additional purification steps (i.e., resin adsorption) that unavoidably increase the environmental and economic impact of the final product [61,62]. The development of a dedicated sustainable approach to separate the eutectic system from the bioactive will be addressed in future work.

#### 3.2.1. Conventional Protocol—ChLA

Table 5 reports the first set of screening, running through the same protocol proposed in Table 1. The ChLA system shows the best activity for prolonged extraction under heating, on the contrary respect to the hydroalcoholic solvent (57.2% Trolox eq. increase vs. 47.5% Trolox eq. decrease, respectively). Thus, it confirmed the stabilising nature of the eutectic medium, which is able to protect the metabolites from thermal and oxidative degradation. The enhancement of product shelf life, already observed on other biomasses, will be investigated in a dedicated paragraph of this manuscript (see Section 3.3.1) [50,59].

#### 3.2.2. Technology Screening—ChLA

As performed for the EtOH/H_2_O system, the ChLA has been tested with a similar approach exploiting UAE and MAE (see Table 6).

The eutectic solvent displayed a different trend again, in contrast with the results collected in Table 3, where, for the hydroalcoholic solvent, the conventional protocol remained the best-performing one. For ChLA compared with MAE, the simple heating protocol, it is possible to appreciate a 49.6% activity increase (as Trolox eq.), whilst UAE achieved a decrease of only 7.8%, in contrast to the 46.7% of EtOH/H_2_O. These results encouraged us to pursue the investigation of the ChLA features for VPP extraction.

#### 3.2.3. US-Assisted Extraction (UAE)—ChLA

The UAE of VPP was further explored, modifying the applied power (up to 500 W) and the extraction time (see Table 7). It is necessary to state that, with respect to the hydroalcoholic solution (see Table 3), it was not possible to increase from 30 to 60 min the process due to the viscous nature of ChLA, which leads to system overheating. For this reason, the extraction protocol was investigated at 15 min.

The UAE carried out at 500 W for 30 min results in the best activity, reaching an overall Trolox eq. of 1874.0 mmol_TE_/g_Extr_, with a 5.7-fold increase if compared with the 15 min at the same power (325.5 mmol_TE_/g_Extr_) and a 6.8-fold increase respect to 100 W (274.5 mmol_TE_/g_Extr_ at 30 min). Further considerations concerning technologies and solvent comparisons will be reported at the end of the next paragraph.

#### 3.2.4. MW-Assisted Extraction (MAE)—ChLA

The VPP MAE with ChLA was performed by retracing the screening protocols proposed in Table 4 for EtOH/H_2_O, increasing the extraction time (60 min vs. 30 min, at 80 °C) and investigating a higher temperature for shorter extraction time (5 min, 120 °C), as well. An intermediate protocol was adopted as a control (30 min, 100 °C). The results are reported in Table 8.

Higher temperatures seem to compensate for the high viscosity of the system, allowing it to enhance the mass transfer. In fact, the extracts achieved at 100 °C and 120 °C exhibit approx. comparable activity (493.7 and 468.2 mmol_TE_/g_Extr_, respectively), even if the latter can count on only 5 min of treatment. However, the comparison between samples at 100 °C and 120 °C suggests that the viscosity reduction is not enough to produce an appreciable activity increment moving from 5 to 30 min, with a consequent kinetic limitation. 

According to the technology screenings reported, the most promising approach with ChLA is proven to be the acoustic cavitation, with a Trolox eq. of 1874.0 mmol_TE_/g_Extr_ (40 °C, 500 W, 30 min), vs. the 493.7 mmol_TE_/g_Extr_ of MAE (120 °C, 5 min) and 510.1 mmol_TE_/g_Extr_ of conventional protocol (80 °C, 4 h). The MW irradiation can assure a very quick protocol able to nearly reach the conventional benchmark in 5 min vs. 4 h. This result can be related to the partial increase in NaDES fluidity (and consequently, mass transfer). On the other hand, extending the treatment to 30 min, the US irradiation exhibits a dramatic extraction intensification. In addition, thanks to the stabilisation features of ChLA, the degradation issues observed for the hydroalcoholic solvent have been avoided.

### 3.3. ChLA Stabilization Effects

It is commonly accepted, due to a massive literature production, that one of the main responsible for the deep eutectic solvent features are the hydrogen bonds, causing their formation and framework arrangement [48,63,64]. Likely, they can help to solubilise a large range of compounds [65], enhance extractions [49], and act in synergy with several actives [66]. Another property ascribed to eutectic solvents is stabilising the molecules dissolved within [50,67]. Thus, we decided to address this point by exploring the VPP/ChLA extracts.

#### 3.3.1. Shelf-Life

The ChLA sample exhibiting the best antioxidant activity (UAE, 30 min, 500 W) has been monitored across 2 years (sampling after 1, 3, 6, 12, 18, and 24 months, respectively, storage at 4 °C), tracking the variation in Trolox equivalents. For the sake of comparison, the same test was performed on the best sample achieved with the hydroalcoholic solvent (MAE, 5 min, 120 °C). The gathered data has been summarised as percentual degradation (normalising on the “0” sample) in Figure 1 and with Pareto charts in Figure 2A,B.

From Figure 1, it is possible to appreciate the degradation trends of the two different solvent systems. The sample in EtOH/H_2_O displayed a higher activity reduction, losing approx. 30% after only 1 month and approaching ca. 70% in 12 months. The final degradation nearly reached 79%. The ChLA system, on the other hand, showed a milder abatement of antioxidant activity across the monitored period, reaching around 57% of the final degradation. To better evaluate the activity variations, the Trolox eq. have been organised by means of Pareto charts (see Figure 2A,B), where the purple line describes the cumulative variation trend.

In Figure 2A, the hydroalcoholic solution is characterised by a steep reduction in Trolox eq., reaching a sort of steady state after 6 months, where the degradation slows down. The antioxidant activity of the eutectic system (Figure 2B), on the contrary, is described by a milder slope without real stationarity even after 24 months.

These results support the enhancement of extracts’ shelf-life according to the antioxidant activity of VPP metabolites, in particular if compared to a common hydroalcoholic medium. It is possible to adopt the antioxidant activity half-life as a reference point, resulting in 4.12 months and 23.22 months for EtOH/H_2_O and ChLA, respectively. Hence, the exploitation of the NaDES granted a 5.6-fold shelf-life extension. To better understand the trend in the shelf-life of the VPPs extracts, preliminary quantification of the total anthocyanin content (TAC) in both the fresh and 24 months EtOH/H_2_O and ChLA extracts have been performed via colourimetric assay (see Section A.1 and Section A.2). Moreover, on ChLA extract a preliminary semi-quantitative anthocyanins detection via LC-MS analysis has been performed recording namely pelargonidin 3-rutinoside-5-glucoside (*m*/*z*: 741.22), pelargonidin 3-feruloylrutinoside-5-glucoside (*m*/*z*: 917.27), and petunidin 3-p-coumaroylrutinoside-5-glucoside (*m*/*z*: 933.27) (see Section A.3 and Section A.4).

#### 3.3.2. Antioxidant Activity Modification: EtOH Addition and US Degradation Tests

After confirming the NaDES stabilisation features, the successive step was to verify how this feature can be affected by the variation of the H-bond strength (see Table 9). Hence, different aliquots of EtOH (1 and 10%) were added to a ChLA extract of “medium activity” (assuring the best range to evaluate both eventual increase and decrease in Trolox equivalents). For this purpose, the sample achieved by means of MW at 100 °C (30 min) was selected.

Table 9 shows an increase in antioxidant activity, dependent on the addition of alcohol, with a maximum of +58.6%. This substantial enhancement is likely dependent on an interference of EtOH with the existing H-bond of ChLA presumably occupied in stabilising the extract bioactive [68,69]. This phenomenon was further studied by submitting the sample “ChLA (+10% *w*/*v* EtOH)” to US irradiation (30 min, as the extraction protocol), evaluating the modification caused by cavitation (see “ChLA Degradation Test (+10% *w*/*v* EtOH)” in Table 9). The choice fell on the acoustic treatment due to the effectiveness exhibited by this technique for VPP extraction. After the US processing, the sample displayed a considerable reduction in antioxidant activity, with a 47.1% loss (from 783.2 to 414.3 mmol_TE_/g_Extr_). It is possible to conclude that reducing the H-bond strength in ChLA results in enhancing the overall extract activity and, as a natural consequence, its proneness to degradation. An additional test was performed to explore this enhancement/destabilisation dualism, evaluating how a small addition of EtOH could effectively influence the quality of a VPP extract achieved by UAE. Table 10 reported the comparison between the best performing US/ChLA extraction (30 min, 500 W) and the same procedure carried out exploiting “ChLA (+1% *w*/*v* EtOH)” as a solvent medium.

As further evidence, the activity of the sample recovered by means of “ChLA (+1% *w*/*v* EtOH)” resulted strongly decreased, more than 77% in comparison to the pure ChLA system (431.3 vs. 1874.0 mmol_TE_/g_Extr_).

A first evaluation of extract composition and how sample aging could affect metabolites are addressed in Appendix A of this manuscript, where some preliminary tests on anthocyanins content and LC-MS analysis are reported.

### 3.4. Biological Activity

During this work, antioxidant features of different VPP extracts have been evaluated, in relation to the two main enabling technologies (MW and US), exploiting a conventional (EtOH/H_2_O) and an unconventional solvent (ChLA, a NaDES)

#### 3.4.1. Antiproliferative Activity

First, the antiproliferative activity of the extracts was investigated on the Caco-2 cell line. This cell line derives from human colorectal adenocarcinoma and is often used as a model for the epithelial barrier of the intestine. Therefore, it was selected to test the extracts obtained for their potential use as a dietary supplement and/or nutraceutical. The bioactive obtained from the VPP could also serve as natural antioxidants for cosmetic products, so the effect of the extracts on the HaCaT cell line was investigated. HaCaT cells are derived from healthy human keratinocytes, which represent 95% of the epidermal cells and primarily have structural and barrier functions in the human skin. The antiproliferative activity was determined by MTS assay. The results are presented in Figure 3.

Figure 3 confirmed that NaDES could enhance the biological activity of the extracts. In particular, extracts prepared in NaDES possess a significantly pronounced antiproliferative effect with respect to the ethanolic extracts. The antiproliferative effect is dose-independent in the case of EtOH and EtOH MW extracts, while both extracts prepared in NADES have shown no statistically significant among doses. This means that even the lowest concentration of NADES extracts ensures exceptional biological activity. Additionally, there is no difference between the effects on the two cell lines employed in this study. The decrease in cell viability is similar on Caco-2 and on HaCaT cells. Due to the observed antiproliferative activity of Ch:LA EtOH, EtOH MW, and Ch:LA extracts, their possible usage as nutraceuticals or in cosmetics could not be expected, at least not in tested volume ratios. Rather, it could be further assessed for possible anticancer activity on different human tumour cell lines, as already shown for different polyphenols extracts obtained by conventional extraction and by green approaches [50,70]. However, keeping in mind the complex relationship between the antioxidant activity of natural compounds and their anticancer activity, supplementation with such extracts requires careful consideration. It is well known that antioxidants may perform a role in cancer prevention by reducing oxidative stress and protecting DNA, but antioxidant supplementation in cancer patients undergoing treatment may interfere with the efficacy of these treatments. Furthermore, different natural compounds with antioxidant properties, obtained as plant extracts, can have diverse effects on cancer cells since they contain multiple bioactive components often characterised by distinct mechanisms of action.

#### 3.4.2. In Vitro Antioxidant Activity

VPP extracts antioxidant activity was evaluated in vitro on both Caco-2 and HaCaT cell lines by DCF-DA assay. The cell lines were treated with 0.5 % *v*/*v* of extracts, the smallest volume ratio, which has a similar impact on cell viability as the higher concentrations. The performed in vitro assay evaluates if the added VPP extracts have antioxidant capacity to prevent ROS (reactive oxygen species) generation during induced oxidative stress in tested cell lines. After pre-treatment with VPP extracts, the oxidative stress was induced by the addition of H_2_O_2_ in the cultivation medium. Lower ROS generation means a higher antioxidant activity of tested extracts. The results are shown in Figure 4.

Pre-treatment with ChLA and ChLA EtOH extracts have shown to decrease ROS generation after the induction of oxidative stress, i.e., can protect cells against the presence of intracellular ROS. Moreover, ChLA extract has shown a significantly higher antioxidant activity when compared to ChLA EtOH extract. Therefore, this extract could be used to prevent oxidative stress in organisms and thus prevent numerous correlated diseases. Other tested extracts did not have such a positive effect; moreover, it seems that the hydroalcoholic extracts fostered the generation of ROS, what is completely opposed to the DPPH· test result.

In particular, according to the DPPH· test (see Table 9), the hydroalcoholic extract achieved by means of MAE should exhibit the best antioxidant activity, but on the contrary, it slightly promoted ROS generation as determined by in vitro DCF-DA assay. The antioxidant activity of ChLA obtained extracts (with and without the EtOH addition) is in good correlation by both applied tests, which confirmed that the eutectic solvent has a higher antioxidant activity than ChLA + EtOH.

The VPP extracts prepared in NaDES showed an antiproliferative effect on the Caco-2 cell line, and HaCaT cell line, with no statistically significant differences among NADES extracts. This effect was much greater than in the extracts prepared in EtOH. Regarding the antioxidative activity, ChLA extract is shown to prevent ROS generation even stronger than ChLA EtOH extract.

Such, in a way, inconsistent results should definitely be further investigated, but it can be explained by a few facts already known in the literature. First, the exact phytochemical profile of the extracts, which was not assessed in this study, obtained by classical extraction and NaDES is definitely different [71]. Otherwise, plant extracts may contain compounds that can either promote or inhibit the formation of ROS, depending on the specific extract, the way of extraction and the conditions of exposure [72]. Finally, the metabolism of ethanol itself can promote the formation of ROS, which could lead to oxidative stress and related cellular damage. Therefore, the effects of (ethanol) plant extracts on ROS formation and oxidative stress are complex and depend on multiple factors, meaning further research is needed to better understand it.

## 4. General Remarks and Perspectives

Recovery of bioactive compounds from food waste not only mitigates environmental problems but also enhances the profitability of the food industry. To achieve this objective, it is necessary to apply innovative technologies that would allow to achieve higher extraction yields, preserving the properties of the recovered extracts. In this framework, the implementation of a biorefinery approach on potato peels as the main source could open future industrial applications that would allow the exhaustive exploitation of this by-product. Recently, a variety of new methodologies for PP management have been successfully applied, such as MAE, UAE, SFE, PEF, or SWE, as reported above. However, in practice, the idiosyncrasy of the agri-food industry and the lack of low-cost industrial equipment have limited the implementation of these technologies. For this reason, easily affordable hydroalcoholic extraction still remains the reference protocol against which each new extraction strategy should be compared, including that herein reported [53,73]. To better clarify the sustainable extraction approach adopted in this paper, a flow chart of the whole VPP valorisation process is reported in Figure 5, together with a comprehensive summary showing the major results recently documented in the literature concerning the antioxidant recovery from PP (Table 11). 

The results documented in this study demonstrate that PP waste could be a good source for antioxidant recovery while contributing to the revalorisation of these agri-food by-products in a sustainable way. When the same violet potato variety was considered for the same purpose, the addition of unsustainable hydrochloric acid was required to stabilise the anthocyanin-rich extract [73], which was not necessary in this work due to the inherent acidity of the ChLA eutectic system used. Despite the potential of the aforementioned food waste and the attention that NADES are receiving in the scientific community, not many studies have been carried out so far concerning the sustainable extraction of phenolic compounds from these residues [18] compared to other biomass sources [77]. However, future research aiming to design effective and easily affordable technological strategies for the complete valorisation of these by-products would be necessary, both for the recovery of bioactive and for their applications.

## 5. Conclusions

The potato is one of the most widely produced vegetables in the world, and due to its extensive use in various industries, large amounts of potato waste are generated. So far, the huge potential of potato waste is underexploited. One of the most promising applications of PPs is the content of active metabolites, which can be extracted using different green technologies, leading to economic and environmental advantages. 

In this study, the exploitation of a deep eutectic solvent, the ChLA, for the valorisation of VPPs proved to be crucial for both the recovery and furtherstabilisationn of target antioxidant compounds.

According to the reported technology screenings (UAE, MAE, and conventional heating), the most promising approach is proven to be acoustic cavitation. Indeed, in the presence of NaDES, UAE enabled 3.6 times higher Trolox equivalents per gram of ChLA extract in 30 min (40 °C, 500 W) than that obtained in 4 h with the hydroalcoholic protocol (1874.0 mmol_TE_/g_Extr_ vs. 510.10 mmol_TE_/g_Extr_, respectively). On the other hand, with the same antioxidant power recorded for the VPP extracts (~500 mmol_TE_/g_Extr_), MAE proved to be effective in drastically shortening the extraction time up to 98%, compared to the conventional hydroalcoholic approach. Moreover, the exploitation of the NaDES granted a 5.6-fold shelf-life extension in terms of its antioxidant activity (monitored over a 24-moths period) if compared to the hydroalcoholic procedure. Finally, the in vitro antiproliferative and antioxidant activity of conventional and ChLA VPP extracts were determined by MTS and DCF-DA assay, respectively, on tumour (Caco-2) and normal human keratinocyte (HaCaT) cell lines. NaDES-extracts exhibited a significantly pronounced antiproliferative effect vs. ethanolic one. They are also the only ones featuring ROS generation decrease after oxidative stress induction in both selected cell lines, confirming that NaDES can enhance the biological activity of the VPP extracts. However, there is a strong need for in vitro and in vivo studies to help better understand the pharmacodynamic and pharmacokinetic properties of these bioactive compounds, helping for the development of new nutraceutical and/or pharmaceutical products as well. This comparative study could pave the way for the development of a synergistic process, combining enabling technologies with green solvents for the extraction of high-added-value products from waste biomass.

## Figures and Tables

**Figure 1 foods-12-02214-f001:**
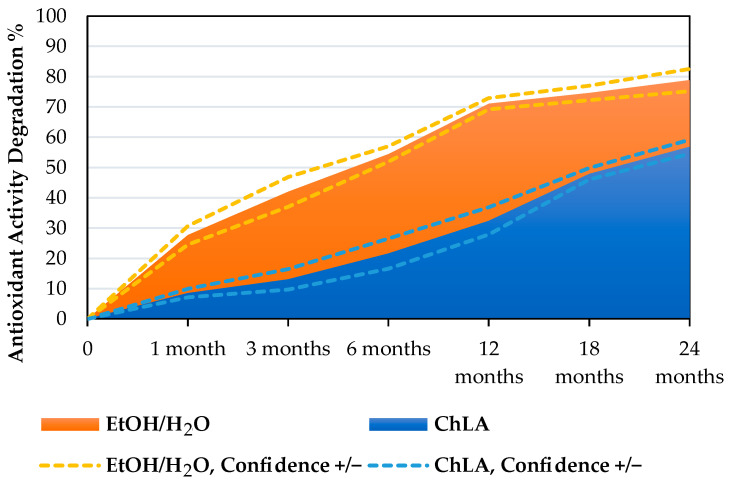
Monitoring antioxidant activity of EtOH/H_2_O vs. ChLA systems over 24 months (degradation). Dashed lines represent confidence intervals resulting from SD.

**Figure 2 foods-12-02214-f002:**
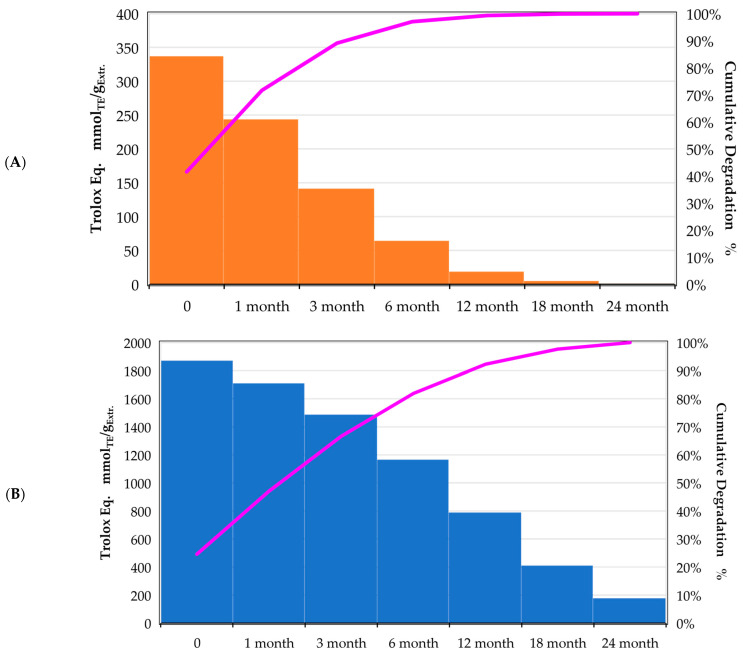
Antioxidant activity reported as Pareto charts, monitored over 24 months; Purple line reports cumulative degradation; (**A**): hydroalcoholic solution; (**B**): ChLA.

**Figure 3 foods-12-02214-f003:**
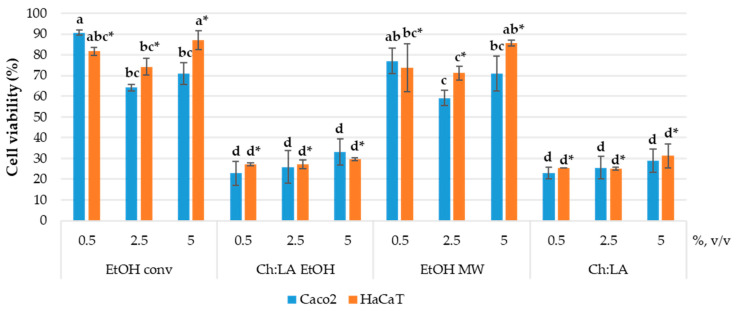
Effect of VPP extracts in the final volume ratios of 0.5, 2.5, and 5% (*v*/*v*) on Caco-2 and HaCaT cell viability. Results are expressed as average values ± SD. Statistically different data (*p* < 0.05) are designated by lower-case letters (a–d or a*–d*, Caco-2 and HaCaT, respectively), according to volume ratios for each cell line.

**Figure 4 foods-12-02214-f004:**
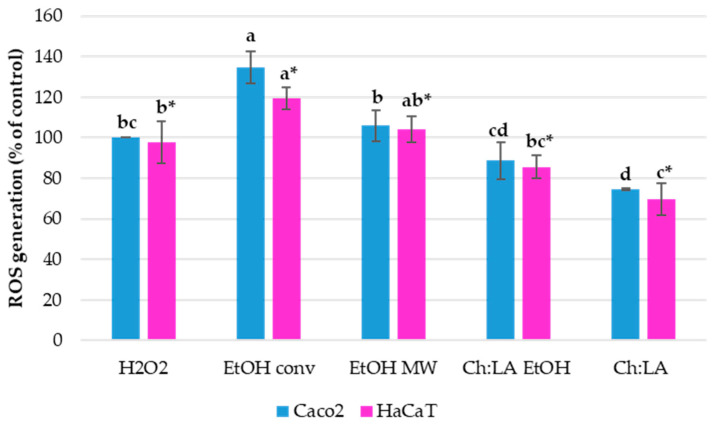
VPP extracts’ effect on intracellular ROS formation in Caco-2 and HaCaT cells during H_2_O_2_-induced oxidative stress. Results are expressed as % of control cells. Results are expressed as average values ± SD. Statistically different data (*p* < 0.05) are designated by lower-case letters (a–d or a*–c*, Caco-2 and HaCaT, respectively).

**Figure 5 foods-12-02214-f005:**
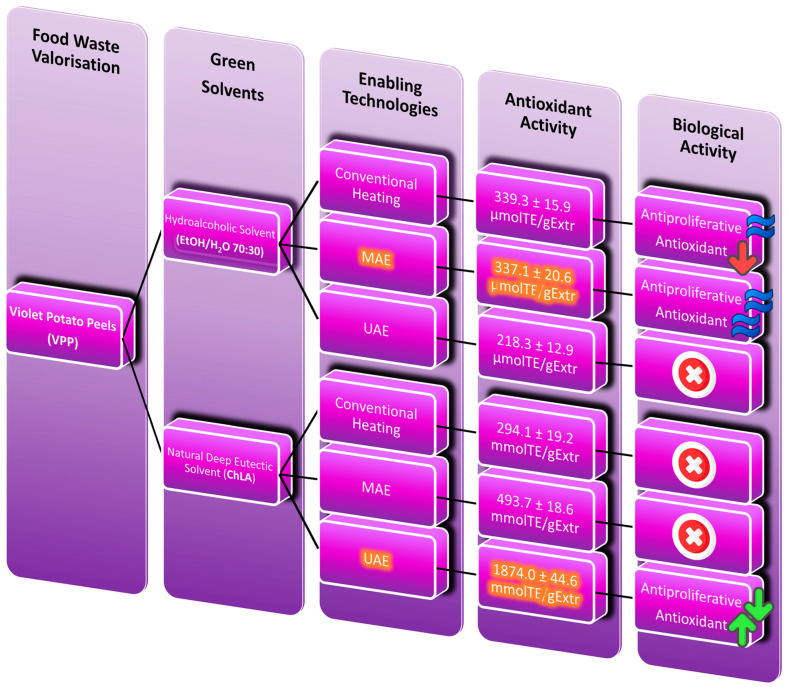
Flowchart of the extraction strategy adopted in this work for the sustainable valorisation of VPP: technologies and solvent screening enabling the best-performing antioxidant activity. Highlighted squares represent the best performing samples.

**Table 1 foods-12-02214-t001:** Results of conventional extraction of violet potato peels in terms of temperature and time screening.

Temperature (°C)	Time (h)	EC_50_ (μg/mL)	Trolox Eq. (µmol_TE_/g_Extr_)	Dry Yield %
80	4	97.7	161.1 ± 8.3	10.36 ± 0.92
RT		66.8	235.7 ± 13.0	13.08 ± 0.88
80	1	46.4	339.3 ± 15.9	13.55 ± 1.12

Extraction conditions: Solvent: EtOH:H_2_O 70:30. Results reported as average value ± S.D.

**Table 2 foods-12-02214-t002:** Results of technological screening for violet potato peel extraction.

Temperature (°C)	Technology	EC_50_ (μg/mL)	Trolox Eq. (µmol_TE_/g_Extr_)	Dry Yield %
80	Conv.	46.4	339.3 ± 15.9	13.55 ± 1.12
	MW	69.4	226.8 ± 12.5	15.67 ± 1.30
40	US (100 W)	87.0	180.9 ± 10.7	14.17 ± 0.99

Extraction conditions: Hydroalcoholic solvent: EtOH:H_2_O 70:30, 30 min; S/L ratio: 1:20. Results reported as average value ± S.D.

**Table 3 foods-12-02214-t003:** Results of cavitational parameters for ultrasound-assisted extraction of violet potato peels.

Time (min)	Power (W)	EC_50_ (μg/mL)	Trolox Eq. (µmol_TE_/g_Extr_)	Dry Yield (%)
30	100	87.0	180.9 ± 10.7	14.17 ± 0.99
60		88.0	178.9 ± 9.8	11.28 ± 1.15
30	500	80.7	195.1 ± 10.8	11.76 ± 1.06
60		72.1	218.3 ± 12.9	10.08 ± 1.57

Extraction conditions: Hydroalcoholic solvent: EtOH:H_2_O 70:30; S/L ratio: 1:20. Results reported as average value ± S.D.

**Table 4 foods-12-02214-t004:** Results of irradiation parameters for microwave-assisted extraction of violet potato peels.

Temperature (°C)	Time (min)	EC_50_ (μg/mL)	Trolox Eq. (µmol_TE_/g_Extr_)	Dry Yield (%)
80	60	69.4	226.8 ± 12.5	15.67 ± 1.30
100	30	48.7	323.2 ± 22.2	4.04 ± 0.86
120	5	46.7	337.1 ± 20.6	10.54 ± 2.01

Extraction conditions: Hydroalcoholic solvent: EtOH:H_2_O 70:30; Extraction time: 30 min; S/L ratio: 1:20. Results reported as average value ± S.D.

**Table 5 foods-12-02214-t005:** Results of conventional NaDES extraction of violet potato peels in terms of temperature and time screening.

Temperature (°C)	Time (h)	EC_50_ (mg/mL)	Trolox Eq. (mmol_TE_/g_Extr_)
80	4	30.8	510.1 ± 42.2
RT		51.7	300.0 ± 18.8
80	1	53.8	292.0 ± 11.7

Extraction conditions: NaDES system: ChLA (1:1 mol/mol). Results reported as average value ± S.D.

**Table 6 foods-12-02214-t006:** Results of technological screening for violet potato peels NaDES extraction.

Temperature (°C)	Technology	EC_50_ (μg/mL)	Trolox Eq. (mmol_TE_/g_Extr_)
80	Conv.	53.8	294.1 ± 19.2
	MW	47.1	330.6 ± 17.9
40	US (100 W)	57.4	274.5 ± 10.8

Extraction conditions: NaDES system: ChLA (1:1 mol/mol); Extraction time: 30 min; S/L ratio: 1:20. Results reported as average value ± S.D.

**Table 7 foods-12-02214-t007:** Results of cavitational parameters for ultrasound-assisted NaDES extraction of violet potato peels.

Time (min)	Power (W)	EC_50_ (mg/mL)	Trolox Eq. (mmol_TE_/g_Extr_)
15	100	74.9	210.1 ± 15.8
30		57.4	274.5 ± 10.8
15	500	48.7	325.5 ± 19.72
30		8.4	1874.0 ± 44.6

Extraction conditions: NaDES system: ChLA (1:1 mol/mol); S/L ratio: 1:20. Results reported as average value ± S.D.

**Table 8 foods-12-02214-t008:** Results of irradiation parameters for microwave-assisted NaDES extraction of violet potato peels.

Temperature (°C)	Time (min)	EC_50_ (mg/mL)	Trolox Eq. (mmol_TE_/g_Extr_)
80	60	47.1	335.1 ± 13.4
100	30	32.2	468.2 ± 22.0
120	5	34.2	493.7 ± 18.6

Extraction conditions: NaDES system: ChLA (1:1 mol/mol); S/L ratio: 1:20. Results reported as average value ± S.D.

**Table 9 foods-12-02214-t009:** Antioxidant activity modification by EtOH addition.

System	EC_50_ (mg/mL)	Trolox Eq. (mmol_TE_/g_Extr_)	Antioxidant Activity Variation (%)
ChLA ^a^	32.2	493.7 ± 18.6	-
ChLA (+1% *w*/*v* EtOH)	24.4	645.2 ± 21.0	+30.7 ^b^
ChLA (+10% *w*/*v* EtOH)	20.1	783.2 ± 23.4	+58.6 ^b^
ChLA Degradation Test (+10% *w*/*v* EtOH) ^c^	38.0	414.3 ± 15.9	−47.1 ^d^

^a^ MAE, 100°C, 30 min; ^b^ Compared to reference extract “ChLA”; ^c^ US 30 min; ^d^ Compared to “ChLA (+10% *w*/*v* EtOH)”. ChLA extract adopted as reference. Results reported as average value ± S.D.

**Table 10 foods-12-02214-t010:** Antioxidant activity modification by US degradation tests. Evaluation of EtOH effect.

Solvent	EC_50_ (mg/mL)	Trolox Eq. (mmol_TE_/g_Extr_)	Antioxidant Activity Variation (%)
ChLA	8.4	1874.0 ± 44.6	-
ChLA (+1% *w*/*v* EtOH)	36.5	431.3 ± 16.7	−77.1

US irradiation parameters: 500 W, 30 min. Results reported as average value ± S.D.

**Table 11 foods-12-02214-t011:** Final comparison between extraction strategies for the recovery of bioactive compounds from potato peels.

Potato Variety	PP Extraction Conditions	Antioxidant Activity (DPPH Assay)	Biological Activity (In Vitro Test)	Ref.
	Conventional extraction			
Agria	71.2% EtOH, 89.9 °C, 34 min	3.2–10.3 mg/100 g db *	Soybean oil stabilisation under accelerated oxidation conditions	[53]
Lady Rosetta	80% MeOH, 23 °C, 15 h	3.51 mg TE/g db	n.d.	[38]
Lady Claire	MeOH and 75% EtOH, 80 °C, 22 min	2.00 mg TE/g db	n.d	[74]
Purple sweet	80% EtOH, acidified by 0.1% (*v*/*v*) HCl 60 °C, 90 min,	1303.14 mg TE/100 g db	n.d.	[73]
Vitelotte	70% EtOH, 80 °C, 1 h	339.3 μmolTE/gExtr	Antiproliferative MTS assay (Caco-2 and HaCaT cell lines)	This work
	Ultrasound-Assisted Extraction (UAE)			
Ratona Morada	70% acetone, 50 °C, 50 min	39.81 EA × 10^−3^ mg TE/g db	n.d.	[75]
Lady Rosetta	80% MeOH, 30–45 °C, 30–900 min, 42–33 Hz	5.86 mg TE/g db	n.d	[38]
Purple sweet	90% EtOH, acidified by 0.1% (*v*/*v*) HCl 50 °C, 45 min, 200 W	1303.14 mg TE/100 g db	n.d.	[73]
Vitelotte	ChLA, 40 °C, 30 min, 500 W	1874.0 mmolTE/gExtr	Antiproliferative MTS assay (Caco-2 & HaCaT cell lines)	This work
	Microwave-Assisted Extraction (MAE)			
Russett Burbank	67.33% MeOH, 15 min, 1:20 S/L ratio,	74 mg TE/g db **	n.d	[36,76]
Vitelotte	ChLA 120 °C, 5 min	493.7 mmolTE/gExtr	Antiproliferative MTS assay (Caco-2 & HaCaT cell lines)	This work
	Pressurised Liquid Extraction (PLE)			
Lady Claire	70% EtOH and 125 °C	3.39 mg TE/g db	n.d.	[74]

n.d.: not documented; d.b.: dry biomass; TE: rolox equivalent; * Total Phenolic Content; ** (%). Pink lines refer to data collected in this work.

## Data Availability

Data is contained within the article.

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
