# Peer review of "Food-Waste Valorisation: Synergistic Effects of Enabling Technologies and Eutectic Solvents on the Recovery of Bioactives from Violet Potato Peels"

_foods, 2023, doi:10.3390/foods12112214_

Round 1
Reviewer 1 Report
The manuscript refers to the combination of enabling technologies and natural eutectic solvents to improve polyphenol extraction from industrial byproducts, in order to enhace the development of circular economy. The aim of the study is intresting, but the manuscript should be modified before further consideration.
- Abstract do not show any results or conclusions. Authors must include main findings of the study.
- Introduction should be shorter, summarized the main concepts needed and show the existin background of this topic. In adittion, references should be cited at the end of the sentence or paragraph to improve understanding.
-Statistical Analysis is not listed in the methods section. Results do not show standard deviation or standard error. Do the different mean values shows significat differences between them? The authors stated differences between treatments but did not show this information.
-Table 1: Why did the authors use dry yield as determinant of extraction success? They also state in table 3 that this is not a reliable marker.
-Line 252: "This information confirms the thermolabile nature of the VPP bioactive fraction". If the bioactive fraction is thermolabile, why do the authors use reflux extraction as a benchmark?
-Authors are trying to use this extracts as anticancer agents? If authors intended to use this extracts as nutraceuticals and as antioxidants in cosmetics, they should not exert any antiproliferative effect in normal cell culture.
-In general, authors do not justify their conclusions or compare their results with existing bibliography related to the topic. References of different articles using standard extraction of polyphenos from potato peels or using enabling technologies and eutectic solvents on other food byproducts should be consulted and added to the manuscript.
Reviewer 2 Report
1. The title needs to be modified as “Food-waste valorization of violet potato peels by exploring the synergistic effects of enabling technologies and eutectic solvents for recovery of phenolic bioactives”.
2. Key quantitative/optimized data should be included in the abstract.
3. The statement of objectives at the end of introduction should be presented as one paragraph and not as points.
4. Section 2.1 and 2.2 should be combined as one subsection.
5. A schematic diagram illustrating the work flow highlighting various extraction methods and activities (total phenolic content, antioxidant activity and antiproliferation) along with key findings for the readers to grasp at a glance.
6. Introduce the word “hydroalcoholic extraction” in the subheading in the section 2 itself instead of latter in the section 3.
7. The number of tables should be reduced by mentioning the text the table contents in the case of very small table (for example Table 1). And several tables can be combined as one figure.
8. All the table and figure captions should be revised in a descriptive way instead of objective way by the use of ‘colon (:)’ symbol. Also, several subsection headings are just objective, which need to be modified.
9. Avoid the unusual usage of the word “furtherly”.
10. For Figure 2A,B, the secondary y-axis is missing.
11. In Tables 2,4,6,7, & 8, there is an unusual horizontal line 2nd, 1st, 2nd, 2nd, 1st and 2nd columns, respectively. What does it represent?
12. All the extraction methods indicated in tables and figures should be described in the respective table footer and figure caption. Likewise, the abbreviations used should be described in full form.
13. A new paragraph just above the conclusion should be included describing the collective discussion of all the optimized parameters and their comparison with the reported data to highlight the superiority of the methods investigated.
14. Conclusion section should be rewritten to summarize the key findings, the limitation of this study and the research gaps identified for future study.
15. The data provided in the Appendix should be briefed in the main text directing the readers to appendix/supplementary for details.
16. In Figures A1-A4, the peaks in the chromatograms should be identified either directly in the chromatogram or in the respective figure caption.
Moderate editing of English language
Reviewer 3 Report
The manuscript of Grillo et al., investigated the extraction of bioactives from violet potato peels using environmentally friendly extraction technologies. The use these innovative technologies is a widely studied topic that is highly relevant. Therefore, this paper deserves recognition in this field of research. The approach of the authors is well performed, the article is well written and the applied methodology has the merits for Foods. However, after reading the manuscript I have minor comments which ought to be addressed before further consideration:
1/ The authors don’t mention stirring speeds for both MAE and UAE. Therefore, were the reactor contents mixed? If this is not the case this could (especially in the case of MAE and UAE) cause (i) local hotspots and subsequent degradation of certain compounds and (ii) mass transfer limitations. Please elaborate in the text.
2/ Did the authors consider (bio)-chemical analysis of the feedstocks that are used in this study (cellulose, hemicellulose, lignin, polyphenolic compounds, fatty acids, etc) ?
3/ What was the power applied in case of UAE? It is later on mentioned in Table 3, but is also important information for the materials and method section.
4/ What was the pore size of filters used in vacuum filtration after MAE and UAE?
5/ The authors mention that all experiments have been conducted in triplicate and that average values are displayed. Would it also be possible to add the standard deviations? That would give an idea about the reproducibility of the experiments.
6/ It is somewhat confusing how the authors refer to the temperature applied in the UAE process. In the materials and methods section it is states: “The mixture was placed in a Pyrex® thimble and cooled by means of an ice bath. The temperature was measured throughout UAE and was maintained under 40 °C to maintain cavitation efficiency.” Later on it is mentioned that: “the US-assisted samples have been collected at temperature of ca. 40 °C.” Could the authors be please more specific towards the reaction conditions applied?
7/ Please insert the equation of “dry yield” is calculated. Is it the residual biomass remaining after above mentioned processes or dry extracted yield in the solvents under scrutiny (based on the tables it would appear the latter). Please also add how this is determined experimentally (for instance via (i) evaporating the solvent and determining via residual mass or (ii) via mass balance through remaining solids in reaction mixture).
8/ Something went wrong in the formatting of Figure 2 (also check Table 11 and Figure 3).
Round 2
Reviewer 1 Report
Authors modified the manuscript and included most of the information that had been requested.
However, authors have not included a statistical analysis. They made conclusions of the results without determining if the means have statistically significant differences.
Example:
Line 269: "Where it is a clearly visible an opposite trend between the antioxidant activity and dry yield"
Taking into account the mean and the standard deviation of dry yield, there would seem to be no significant differences.
Line 510: Figure 3 confirmed that NaDES can enhance the biological activity of the extracts. In particular, extracts prepared in NaDES possess a significantly pronounced antiproliferative effect in respect to the ethanolic extracts. Their effect is dose independent. Also, there is no significant difference between effects on the two cell lines employed in this study.
Line 541: Pretreatment with ChLA extract is the only one that decreases ROS generation after 541 induction of oxidative stress
Reviewer 2 Report
The authors have satisfactorily addressed all the comments raised by reviewers and therefore I recommend acceptance of this article for publication in Foods.
Minor editing of English language required
